# Helicopter Emergency Medical Service (HEMS) Response in Rural Areas in Poland: Retrospective Study

**DOI:** 10.3390/ijerph16091532

**Published:** 2019-04-30

**Authors:** Patryk Rzońca, Stanisław Paweł Świeżewski, Rakesh Jalali, Joanna Gotlib, Robert Gałązkowski

**Affiliations:** 1Department of Emergency Medicine, Faculty of Health Sciences, Medical University of Lublin, 4-6 Staszica St., 20-081 Lublin, Poland; 2Department of Emergency Medical Services, Faculty of Health Science, Medical University of Warsaw, 81 Żwirki i Wigury St., 02-091 Warsaw, Poland; stanislaw.swiezewski@gmail.com (S.P.Ś.); r.galazkowski@lpr.com.pl (R.G.); 3Department of Emergency Medicine, School of Medicine, Collegium Medicum University of Warmia and Mazury in Olsztyn, 30 Aleja Warszawska St., 10-082 Olsztyn, Poland; rakesh.jalali@uwm.edu.pl; 4Division of Teaching and Outcomes of Education, Faculty of Health Science, Medical University of Warsaw, 81 Żwirki i Wigury St., 02-091 Warsaw, Poland; joanna.gotlib@wum.edu.pl

**Keywords:** Helicopter Emergency Medical Service, prehospital care, rural region, health of rural populations, sudden cardiac arrest, acute coronary syndromes, cardiovascular diseases

## Abstract

The aim of the study was to identify the characteristics of missions performed by HEMS (Helicopter Emergency Medical Service) crews and the analysis of health problems, which are the most common cause of intervention in rural areas in Poland. The study was conducted using a retrospective analysis based on the medical records of patients provided by the HEMS crew, who were present for the emergencies in rural areas in the period from January 2011 to December 2018. The final analysis included 37,085 cases of intervention by HEMS crews, which accounted for 54.91% of all the missions carried out in the study period. The majority (67.4%) of patients rescued were male, and just under a quarter of those rescued were aged between 50–64 years. Injuries (51.04%) and cardiovascular diseases (36.49%) were the main diagnoses found in the study group. Whereas injuries were significantly higher in the male group and patients below 64 years of age, cardiovascular diseases were higher in women and elderly patients (*p* < 0.001). Moreover, in the group of women myocardial infarction was significantly more frequent (30.95%) than men, while in the group of men head injuries (27.10%), multiple and multi-organ injuries (25.93%), sudden cardiac arrest (14.52%), stroke (12.19%), and epilepsy (4.95%) was significantly higher. Factors that are associated with the most common health problems of rural patients are: gender and age, as well as the seasons of the year and the values of the Glasgow Coma Scale (GCS), Revised Trauma Score (RTS), and National Advisory Committee for Aeronautics (NACA) used to assess the clinical status of patients.

## 1. Introduction

Despite the constant development and progress made in medicine, mortality resulting from various causes, such as cardio vascular diseases, cerebral disorders, and life-threatening injuries, is a huge problem and challenge for health care systems around the world. According to the data from the World Health Organization (WHO), 56.9 million deaths were reported worldwide in 2016, more than half of which were caused by 10 major causes, including ischemic heart disease, stroke, chronic obstructive pulmonary disease, or lower respiratory tract infections [1,2].

Ensuring health security by guaranteeing fair and equal access to medical services, regardless of health needs, social and economic conditions, or place of residence is one of the priority tasks of administrative and governmental authorities of every country in the world. Although living in rural areas should not affect the availability of health care services, according to research conducted by WHO, there are significant disparities in terms of access to health care between urban and rural areas across the world [3,4]. The problem of inequalities in access to medical care resulting from the place of residence has been confirmed in studies, which show that rural residents rarely get the privileges for regular medical appointments, prophylactic programs, and have much weaker access to emergency services, which are more often located within urban areas. Prophylactic programs, including the early diagnosis of civilization diseases and better access to medical services, should be implemented to eliminate the problem of the differences in the health state of rural and urban residents. Only such a health policy will ensure sustainable development in rural areas [5,6,7,8,9].

The Helicopter Emergency Medical Service (HEMS) has been established and operates in many countries as one of the elements of the health care system that provide care to people in immediate life-threatening situations. The availability of HEMS is important both in rural areas as well as in places difficult to reach for ground-based emergency teams. In Poland, the medical rescue system consists of two important elements, Hospital Emergency Departments (ED) and Medical Rescue Teams (EMS), along with Air Rescue Teams (HEMS).

The responsibility of pre-hospital patient management and transport to the hospital lies in hands of EMS and HEMS crews. HEMS crews in Poland are part of the Medical Air Rescue Service. Air Rescue crews are the only medical service using rescue helicopters as a means of transport for patient support. In Poland there are 21 permanent bases and one seasonal (from June to September) that provide medical care throughout the country (38 million inhabitants). The HEMS crew consists of a professional pilot, paramedic/nurse, and a doctor.

The medical dispatcher directs the closest available helicopter to the scene of the event, whose readiness for take-off during the day depends on the distance of the HEMS team’s location from the scene of the event. It is usually 3 min from the notification within a radius of up to 60 km, 6 min from the notification within the radius from 60 to 130 km, or 15 min after the notification within the radius of more than 130 km. The distance dependent readiness for take-off is directly related to the amount of fuel to be tanked in order to maintain the first-class performance of the rescue helicopter. All helicopters are equipped with proper medical equipment and medicines enabling advanced medical rescue operations, including pre-hospital post-resuscitation care [10,11].

The present scenario of health care problems in rural areas prompted the authors to undertake the research aimed at identifying and presenting the characteristics of missions performed by HEMS crew and analysis of health problems, which are most often the cause of responses in rural areas in Poland.

## 2. Materials and Methods

### 2.1. Study Design

The study is a retrospective analysis of interventions carried out by crews of the Helicopter Emergency Medical Air Rescue Service. All missions of the involvement of the HEMS crew members, from the period from January 2011 to December 2018, in rural areas were included in the study except for the cases where the HEMS crew activation was cancelled, the mission was interrupted due to poor meteorological conditions, the patient refused treatment, and lack of patient on site, which were excluded from the study. Hence 37,085 cases of intervention by HEMS crew were qualified for final analysis that represented 54.91% of all missions carried out in the period of study. The research project did not require the consent of the Bioethics Committee.

The study was carried out on the basis of medical documentation of patients treated by HEMS crew with prior approval of the General Director of the HEMS. The data obtained for the study was gender and age of patients, season and the year of intervention, parameters of the clinical status of the patients, medical procedures undertaken during the mission, main diagnosis according to ICD- 10 classification, and the information related to the characteristics of HEMS mission such as duration of the mission, distance to the place of the event and transport to the hospital, type of mission, and the required treatment of the patient. The clinical status was assessed on the basis of clinical symptoms and three scales: Glasgow Coma Scale (GCS), Revised Trauma Score (RTS), and National Advisory Committee for Aeronautics (NACA).

Glasgow Coma Scale (GCS) is widely used for assessing patient awareness by both pre-hospital Emergency Medical Teams and hospital staff. The GCS score consists of three components: visual, verbal, and motor. The visual component is a score from 1 to 4 points, the verbal component a sore from 1 to 5 points, and the motor component a score from 1 to 6 points. The maximum number of points for the GCS scale is 15 points while the minimum is 3 points. GCS values can be divided into three categories of consciousness disorders: severe (GCS 3–8), moderate (GCS 9–12), and mild (GCS 13–15) [12,13,14].

Revised Trauma Score (RTS) is the scale used in pre-hospital set up to assess the severity of injuries. It is the weighted sum of the coded variables of Glasgow Coma Scale (GCS), systolic blood pressure (SBP), and respiratory rate (RR). Each variable is scored on a scale of 0 to 4 points, with a maximum of 12 points and a minimum of 0 points. The higher the score the higher the chance of survival. RTS < 5 requires transfer to the trauma centre [14,15].

The National Advisory Committee for Aeronautics (NACA) is a commonly used scale for prehospital severity assessment of injury or disease in Western European countries. This scale distinguishes eight groups of patients according to the severity of vital impairment due to injury, illness or poisoning where:–NACA 0 means no injury or disease;–NACA 1 means non-medical injury or illness;–NACA 2 means injury or illness requiring investigation and treatment, but without the need for hospitalization;–NACA 3 means injury or disease without an acute life-threatening situation but requires hospitalization;–NACA 4 means injuries or illnesses that may lead to a deterioration in vital signs;–NACA 5 means injury or illness that is acute life-threatening;–NACA 6 means bodily injury or illness leading to sudden cardiac arrest;–NACA 7 means death [16,17].

### 2.2. Statistical Analysis

The data obtained from the medical records were subjected to statistical analysis using STATISTICA 13 program (StatSoft, Cracow, Poland). In the description of the qualitative data, the number (*n*) and percentage (%) were used, while in the numerical data—the mean (M) and Standard Deviation (SD).

To check the normality of the distribution of quantitative variables, the Shapiro-Wolf normality test was used. The Chi^2^ test was used to assess statistically significant differences between quality variables. The Kruskal-Wallis non-parametric test was used to examine the differences between many groups. The significance level was assumed to be *p* < 0.05.

## 3. Results

Table 1 presents a detailed description of the examined patients and their clinical condition. The majority (67.4%) of patients rescued were male, and just under a quarter of those rescued were aged between 50–64 years. The average age for the whole study group was 48.87 years (SD 24.03). The main diagnosis that led to HEMS interventions in rural areas were injuries, which accounted for more than half of all cases (51.04%), and the most common were the head injuries (25.94%). The analysis of the clinical status of the patients revealed an average GCS scale of 12.07 (SD 4.50), RTS 10.20 (SD 3.67), and NACA 4.04 (SD 1.49).

Table 2 shows the characteristics of the HEMS missions carried out in rural areas. Most interventions were implemented in 2018 (16.20%) and more than a third of HEMS calls were in the summer period (37.61%). More than half of all interventions were sent to accidental events (54.85%). HEMS crew were most often called to assist the ground teams (75.18%), and later involved in patient transport to the hospital (83.32%). The most common medical rescue procedures were spinal board immobilization (33.69%), oxygen therapy (29.16%), and cervical spine stabilization (28.77%). In the records of study material, the mean time of arrival at the site of the event was 16.64 min (SD 6.88), with an average distance of 42.15 km (SD 21.92) to the event site. The mean time from the dispatch of the HEMS crew to the patient’s arrival was 19.08 min (SD 7.40) and the on-site treatment time was 19.39 min (SD 10.27). The mean transportation to hospital was 15.33 min (SD 7.07) at an average distance of 40.72 km (SD 22.48).

The statistical analysis showed a statistically significant relationship between the most common diagnoses according to ICD-10 classification and the gender and age of patients, the season of the year, the GCS, the RTS, the NACA scores, and patient death (*p* < 0.0001). As shown in Table 3, cardiovascular diseases (42.83%), nervous system disorders (5.17%), respiratory diseases (2.43%), and other diagnoses (3.23%) were significantly more common in the group of women. In the group of men, injuries (54.91%) were more frequent. The analysis shows that younger patients were more frequently diagnosed with injuries (37.28 years) and external causes of illness and death (37.15 years), while in older patients cardiovascular (65.72 years) and respiratory (57.73 years) diseases were more significant (*p* < 0.0001). The analyzed material revealed that injuries (55.03%) dominated in summer and cardiovascular diseases (43.97%), nervous system diseases (5.48%), respiratory diseases (2.79%), and other diagnoses (3.06%) were mainly noticed in winter. The analysis of the clinical status of the examined patients shows that the patients with cardiovascular impairment appeared mostly in the severe condition, as confirmed by GCS (10.90), RTS (9.08), and NACA (4.36) scales (*p* < 0.0001). Cardiovascular diseases (15.53%) and external causes of morbidity and death (10.51%) had the highest percentage of deaths among patients treated by HEMS crews in rural areas.

The statistical analysis showed a statistically significant relationship between the most common health problems and the gender and age of patients, the season of the year, the GCS, the RTS, the NACA scores, and patient death (*p* < 0.0001). An analysis of the most common health problems of rural patients shows that acute coronary syndromes (ACS) were significantly more frequent in women (30.95%), while head injuries (27.10%), multiple and multi-organ injuries (25.93%), sudden cardiac arrest (SCA) (14.52%), stroke (12.19%), and epilepsy (4.95%) were more common in the male group. Head injuries (35.71 years) and multiple and multi-organ injuries (38.12 years) were more common in younger patients, while ACS (72.50 years) and stroke (64.87 years) were common in elderly patients (*p* < 0.0001). The analysis shows that ACS (24.97%), SCA (15.25%), stroke (14.60%), and epilepsy (6.03%) were more frequent in winter, while head injuries (27.68%) and multiple and multi-organ injuries (26.45%) occurred mostly in summer. The most severe clinical condition in the study group was associated with patients suffering from sudden cardiac arrest and multiple and multi-organ injuries, as confirmed by GCS (3.59; 11.47), RTS (2.05; 9.84) and NACA (5.79; 4.49) (*p* < 0.0001). The highest percentage of patient deaths (48.41%) was recorded in the case of cardiac arrest calls. Table 4 presents the detailed analysis.

## 4. Discussion

According to WHO data, ischemic heart disease and stroke in 2016 led to 15.2 million deaths and are the biggest killers in the world, as they have remained the main causes of death in the last decade [1,2]. Ensuring public health care security is the priority of all health care systems worldwide. Despite continuous developments in medicine and increasing financial resources for medical care, many countries are still facing problems of unequal access to health care, especially in rural areas [5,7,8]. The main reason behind the improper access to health care in rural areas is the distance to health care facilities. The solution found by many countries to eliminate this problem is the establishment of the Helicopter Emergency Medical Rescue Service, which has made it possible to reduce the time of arrival to the event site, that is far from health care facilities, as well as the time of patient arrival at the hospital [11,18]. The issues of accessibility and actions taken to improve the quality of medical services in rural areas prompted the authors of this paper to present the characteristics of missions performed by HEMS crew and the analysis of health problems, which are the most cause of the HEMS interventions in rural areas. This is the first retrospective analysis carried out in Poland regarding the interventions by HEMS crew in the rural areas covering 37,085 cases, which accounted for over half of all the missions carried out in the analyzed period. Similar results were obtained by Raatiniemi et al. (2015), who analyzed cases of patients with injuries seen by Finnish HEMS [19]. On the other hand, studies by Goldstein et al. (2015) and Newgard et al. (2017) concerning the analysis of interventions by EMS showed that more frequent medical rescue activities were carried out in urban areas [20,21].

In this study, the number of male patients, and patients aged 50–64 years with an average age of 48.87 years was significantly higher. Similar demographic features were obtained by other authors analyzing interventions by helicopter medical rescue service crew—Østerås et al. (2017), Kottmann et al. (2018), Norum and Elsbak (2011) [22,23,24]. Although the studies by Mathiesen et al. (2018), McMullan et al. (2012), Baker and McKay (2010), and Christensen et al. (2017) also showed that male group patients was significantly higher in number, the mean age of patients in their study was higher than identified in this report [25,26,27,28].

Our findings showed that injuries and cardiovascular diseases were the most common diagnosis that led to HEMS interventions. The main health problems in the study group with injuries were head injuries and multiple and multi-organ injuries. Carron et al. (2015) demonstrated in their studies that the main causes of emergency intervention were injuries, respiratory failure, and cardiovascular problems [29]. Wang et al. (2013) assessed the characteristics of emergency services in the United States and found that the predominant causes of emergency calls were body injuries, consciousness disorders, fainting, convulsions, and hypoglycemia, followed by chest pain and heart disease [30]. Kornhall et al. (2018) in a study on HEMS interventions in the rural areas in Sweden showed that injuries, chest pain, and SCA were the main causes of HEMS crew disposal [31]. Similar results were obtained by Østerås et al. (2017) in a study on HEMS interventions in Norway [22].

The literature highlights the aspect of the systematic increase in demand for airborne support in the context of accidents and sudden illnesses in sparsely populated areas. Rapid activation of HEMS crews and the possibility of transporting the injured to distant centres with appropriate references are used in the therapeutic process of patients from rural and difficult-to-reach areas. [11,18,32]. The results of our own research allow us to observe a relatively stable demand for HEMS interventions for the population living in rural areas in the years 2011–2016, and then a significant increase in these missions from 2017.

In the further part of the paper, we analyzed the causes and health problems of the rural population that led to interventions by the HEMS crew. The results of our own research showed that injuries were the main problems, and the dominant health problems were head injuries and multiple and multi-organ injuries. It should be emphasized that the problem of injuries resulting from falls, road traffic accidents, and burns is a huge challenge for the public health sector and is the subject of numerous studies [33,34,35,36,37,38]. The analysis by Fazel et al. (2012) of the profile of adults injured in Iran found that injuries mainly occurred in men, people aged 33.18 (SD 10.90) years on average, and the further management of the injured person required the transport to hospital [33]. Huang et al. (2016) in their studies described patients injured as a result of road accidents and transported by ambulance. Their results showed that the victims transported by the ambulance are more often men, people aged 50–59 years [35]. On the other hand, Bigdeli et al. (2010) in the study on pre-hospital care of road accident victims in Iran showed that in cities and on national highways traffic accidents are more frequent in spring [37]. In turn, Rzońca et al. (2017) conducted research on the use of HEMS for pedestrian injuries and found that HEMS crew were more often dispatched in summer and the victims were mainly men, people under 44 years of age, who were diagnosed with multi-organ injuries, and transported to the hospital [38]. The results of our own research showed that HEMS intervened in rural areas for head injuries and multiple and multi-organ injuries mainly in men and younger population during summer.

The analysis of our own research shows that cardiovascular diseases were the second main diagnosis. The main cardiovascular diseases were acute coronary syndromes and stroke. Acute coronary syndrome was more frequently diagnosed in women, whereas SCA and stroke was more common in men. Studies reveal that in many countries in the world men are more likely to have worse health and higher mortality, due to numerous conditions, including increased cardiovascular risk, as confirmed in the studies by Hess et al. (2007) and Palomo et al. (2014) [39,40]. The increased risk of cardiovascular diseases among women is emphasized by Mosca et al. (2011) and Nahhas et al. (2014) [41,42]. Hess et al. (2007), Schewe et al. (2015), Cebula et al. (2016), El Sayed el at. (2017) in their studies on out of hospital cardiac arrest (OHCA) indicate that OHCA is more common in men. [39,43,44,45]. Hess et al. (2007), Schewe et al. (2015), Cebula et al. (2017), El Sayed et al. (2017), Hawkes et al. (2017), in their studies on out-of-hospital cardiac arrest, one of the greatest challenges for health care systems around the world, have shown that the average age of people with OHCA (Out of Hospital Cardiac Arrest) is just over 60 years [39,43,44,45,46].

Our study reveals that ACS (72.50 years), stroke (64.87 years), and SCA (59.86 years) were more frequent in older patients from rural areas with HEMS crew at their disposal. Marti-Soler et al. (2014) showed a seasonal character of cardiovascular risk factors - higher in winter and lower in summer, which in turn can be defined as the seasonal mortality due to cardiovascular diseases [47]. Our study confirms the above, as our results show that ACS, SCA, and stroke were more frequently diagnosed in winter.

The results of our own research showed that the clinical status of patients seen by HEMS crew varied. The average GCS score was 12.07 and RTS 10.20, while NACA 4.04, which indicates that patients in the analyzed material were most often injured or ill, which could lead to the deterioration, but were not directly life-threatening situations. The analysis shows that the patients suffering from cardiovascular diseases, SCA, and multiple and multi-organ injuries were in most severe clinical condition. Starnes et al. (2018) compared the condition of trauma patients transported by ground and air medical rescue teams from rural areas and revealed that the average GCS score for HEMS was 11.3, and for ground medical teams it was 13.3 [48]. In a study regarding intervention by HEMS in the rural areas in Sweden [22] and Norway [31], the mean NACA score was 4, with the highest values for SCA and chest pain. However, a study by Kottmann et al. (2018) showed that the clinical status of patients with injuries seen by HEMS crew in Switzerland was assessed as mild to moderate (NACA 1–3) [23].

Our research is the first Polish retrospective analysis of response by HEMS crew in rural areas with a significant number of cases analyzed covering all HEMS bases of the Polish Medical Air Rescue Service has some limitations. The analysis covers only pre-hospital operations and the assessment of clinical condition on the basis of information available in the medical records of HEMS crew, without the knowledge of further diagnosis and tests carried out in the in-hospital setup. This makes it impossible to trace the entire therapeutic process of patients. However, these limitations do not affect the quality of the performed study. It is necessary to conduct further research on health care in rural areas in order to gain a better understanding of this issue and at the same time ensure the best possible quality of medical services by entities that provide care to patients living in rural areas, both in pre-hospital and in-hospital settings.

## 5. Conclusions

The most common causes of HEMS interventions in rural areas were injuries and cardiovascular diseases, of which the most common health problems for patients living in rural areas were head injuries and multiple and multi-organ injuries. Interventions by HEMS crew found that the major group was men and people aged 50–64 years.

Factors that are associated with the most common health problems of rural patients are: gender and age of patients, as well as the seasons of the year, and the values of scale parameters for the assessment of clinical condition of patients. Injuries were significantly more frequently diagnosed in the group of men and younger patients, while cardiovascular diseases were diagnosed mainly in women and elderly patients, which is also confirmed by global trends in morbidity and mortality.

## Figures and Tables

**Table 1 ijerph-16-01532-t001:** Patient characteristics and clinical status.

**Gender—*n* (%)**	
Female	11968 (32.36)
Male	25012 (67.64)
**Age in years *n* (%)**	
<19	4745 (13.54)
19–34	6057 (17.28)
35–49	5479 (15.63)
50–64	8313 (23.71)
65–79	6620 (18.89)
≥80	3840 (10.95)
**Age (years)—M (SD)**	48.87 (24.03)
**Main Diagnosis—*n* (%)**	
Injuries	18856 (51.04)
Cardiovascular diseases	13482 (36.49)
Neurological disorders	1626 (4.40)
External cause of disease and death	1379 (3.73)
Respiratory diseases	708 (1.92)
Other diagnosis	892 (2.41)
**Common health problems—*n* (%)**	
Head injury	5075 (25.94)
Multiple and multi-organ injury	4678 (23.91)
ACS	3981 (20.35)
SCA	2553 (13.05)
Stroke	2326 (11.89)
Epilepsy	948 (4.85)
**GCS M(SD)**	12.07 (4.50)
≤8	7389 (21.69)
9–12	3139 (9.22)
≥13	23534 (69.09)
**RTS M(SD)**	10.20 (3.67)
**NACA M(SD)**	4.04 (1.49)

ACS—Acute Coronary Syndromes; SCA—Sudden Cardiac Arrest; GCS—Glasgow Coma Scale; RTS—Revised Trauma Score; NACA—National Advisory Committee for Aeronautics.

**Table 2 ijerph-16-01532-t002:** Helicopter Emergency Medical Service (HEMS) Mission Characteristics.

**Year of Intervention—*n* (%)**	
2011	4410 (11.93)
2012	4084 (11.04)
2013	4247 (11.48)
2014	4630 (12.52)
2015	4658 (12.60)
2016	38.63 (10.45)
2017	5098 (13.79)
2018	5990 (16.20)
**Season of year—*n* (%)**	
Winter	4912 (13.28)
Spring	12026 (32.52)
Summer	13909 (37.61)
Autumn	6133 (16.58)
**Type of mission—*n* (%)**	
Accident	20282 (54.85)
Disease	16698 (45.15)
**Patient care—*n* (%)**	
HEMS transport to hospital	30812 (83.32)
Ambulance transport to hospital	2027 (5.48)
Declared dead	2905 (7.86)
Patient discharged on site	1236 (3.34)
**Assist the ground team—*n* (%)**	
Yes	27801 (75.18)
No	9179 (24.82)
**Procedures carried by HEMS—*n* (%)**	
Spinal board immobilization	12460 (33.69)
Oxygen therapy	10784 (29.16)
C-spine stabilization—Collar placement	10638 (28.77)
i.v. access	8604 (23.27)
Trauma care	6708 (18.14)
Mechanical ventilation	5710 (15.44)
RSI	4932 (13.34)
On scene arrival time (min) M (SD)	16.64 (6.88)
Time from takeoff to on scene arrival to patient (min) M (SD)	19.08 (7.40)
Time of on scene intervention (min) M (SD)	19.39 (10.27)
Hospital transport time (min) M (SD)	15.33 (7.07)
Distance to the site of emergency *(km)* M (SD)	42.15 (21.92)
Distance to the hospital (km) M (SD)	40.72 (22.48)

RSI—Rapid Sequence Intubation.

**Table 3 ijerph-16-01532-t003:** Statistical analysis of most common diagnosis in correlation with patient characteristics and clinical status.

Variables	Common Diagnosis ICD—10						*p*-Value
	Injuries	Cardiovascular Diseases	Neurological Disorders	External Cause of Disease and Death	Respiratory Diseases	Other Diagnosis	
**Gender**—*n* (%)							<0.0001 a
Female	5133 (42.96)	5118 (42.83)	618 (5.17)	404 (3.38)	290 (2.43)	386 (3.23)	
Male	13723 (54.91)	8364 (33.46)	1008 (4.03)	975 (3.90)	418 (1.67)	506 (2.02)	
**Age (years)***—n* (%)							<0.0001 a
<19	3626 (76.47)	261 (5.50)	370 (7.80)	265 (5.59)	107 (2.26)	113 (2.38)	
19–34	4908 (81.07)	429 (7.09)	204 (3.37)	354 (5.85)	35 (0.58)	124 (2.05)	
35–49	3679 (67.22)	1077 (19.68)	249 (4.55)	285 (5.21)	40 (0.73)	143 (2.61)	
50–64	3511 (42.26)	3918 (47.15)	352 (4.24)	238 (2.86)	114 (1.37)	176 (2.12)	
65–79	1481 (22.39)	4390 (66.37)	250 (3.78)	115 (1.74)	199 (3.01)	179 (2.71)	
≥80	409 (10.65)	3006 (78.28)	127 (3.31)	28 (0.73)	155 (4.04)	115 (2.99)	
**Age***—*M (SD)	37.28 (20.78)	65.72 (16.76)	43.50 (25.76)	37.15 (20.84)	57.73 (27.28)	50.76 (24.76)	<0.0001 b
**Season of year**—*n* (%)							<0.0001 a
Winter	2038 (41.52)	2158 (43.97)	269 (5.48)	156 (3.18)	137 (2.79)	150 (3.06)	
Spring	6051 (50.36)	4455 (37.08)	523 (4.35)	449 (3.74)	251 (2.09)	287 (2.39)	
Summer	7646 (55.03)	4562 (32.83)	545 (3.92)	603 (4.34)	202 (1.45)	336 (2.42)	
Autumn	3121 (50.96)	2307 (37.67)	289 (4.72)	171 (2.79)	118 (1.93)	119 (1.94)	
**GCS**—M (SD)	12.81 (4.00)	10.90 (5.05)	12.26 (3.60)	11.98 (4.72)	12.70 (3.97)	13.48 (3.15)	<0.0001 b
≤8	2912 (39.44)	3735 (50.58)	253 (3.43)	300 (4.06)	106 (4.06)	78 (1.06)	<0.0001 a
9–12	1099 (35.01)	1543 (49.16)	266 (8.47)	78 (2.48)	65 (2.07)	88 (2.80)	
≥13	13394 (59.95)	7149 (30.40)	928 (3.95)	916 (3.89)	460 (1.96)	673 (2.86)	
**RTS**—M (SD)	10.85 (2.79)	9.08 (4.63)	11.23 (1.44)	9.96 (4.02)	10.95 (2.18)	11.29 (2.09)	<0.0001 b
**NACA**—M (SD)	3.93 (1.35)	4.36 (1.58)	3.52 (1.17)	3.78 (1.73)	3.73 (1.45)	3.00 (1.48)	<0.0001 b
**Patient death**—*n* (%)	676 (3.59)	2094 (15.53)	1 (0.06)	145 (10.51)	8 (1.13)	23 (2.58)	<0.0001 a

a—Chi^2^ test; b—Kruskal-Wallis test; GCS—Glasgow Coma Scale; RTS—Revised Trauma Score; NACA—National Advisory Committee for Aeronautics.

**Table 4 ijerph-16-01532-t004:** Statistical analysis of most common diseases in correlation with patient characteristics and clinical status.

Variables	Most Common Health Problems						*p*-Value
	Head Injury	Multiple and Multiorgan Injuries	ACS	SCA	Stroke	Epilepsy	
**Gender**—*n* (%)							<0.0001 a
Female	1485 (23.51)	1243 (19.68)	1955 (30.95)	630 (9.97)	711 (11.26)	292 (4.62)	
Male	3590 (27.10)	3435 (25.93)	2026 (15.30)	1923 (14.52)	1615 (12.19)	656 (4.95)	
**Age (years)***—n* (%)							<0.0001 a
<19	1265 (56.45)	610 (27.22)	8 (0.36)	95 (4.24)	0 (0.00)	263 (11.74)	
19–34	1175 (40.27)	1425 (48.83)	23 (0.79)	141 (4.83)	11 (0.38)	143 (4.90)	
35–49	800 (31.34)	954 (37.37)	151 (5.91)	276 (10.81)	196 (7.68)	176 (6.89)	
50–64	864 (18.99)	839 (18.44)	837 (18.40)	833 (18.31)	973 (21.39)	203 (4.46)	
65–79	417 (10.71)	348 (8.94)	1518 (38.99)	709 (18.21)	802 (20.60)	99 (2.54)	
≥80	127 (5.63)	83 (3.68)	1386 (61.49)	322 (14.29)	314 (13.93)	22 (0.98)	
**Age***—*M (SD)	35.71 (22.20)	38.12 (19.04)	72.50 (12.80)	59.86 (18.45)	64.87 (11.83)	50.93 (23.91)	<0.0001 b
**Season of year**—*n* (%)							<0.0001 a
Winter	568 (21.82)	451 (17.33)	560 (24.97)	397 (15.25)	380 (14.60)	157 (6.03)	
Spring	1638 (26.04)	1485 (23.61)	1324 (21.05)	809 (12.86)	740 (11.76)	294 (4.67)	
Summer	2009 (27.68)	1920 (26.45)	1296 (17.85)	905 (12.47)	799 (11.01)	330 (4.55)	
Autumn	860 (25.23)	822 (24.11)	711 (20.86)	442 (12.97)	407 (11.94)	167 (4.90)	
**GCS**—M (SD)	11.74 (4.43)	11.47 (4.74)	11.94 (3.29)	3.59 (2.26)	14.32 (2.44)	12.65 (3.14)	<0.0001 b
≤ 8	1203 (22.97)	1141 (21.78)	536 (10.23)	2147 (40.99)	105 (2.00)	106 (0.02)	<0.0001 a
9–12	416 (18.40)	374 (16.54)	1224 (54.14)	26 (1.15)	49 (2.17)	172 (7.61)	
≥ 13	3006 (29.24)	2691 (26.18)	1959 (18.99)	74 (0.72)	1986 (19.32)	570 (5.55)	
**RTS**—M (SD)	10.62 (2.65)	9.84 (3.77)	11.21 (1.29)	2.05 (3.33)	11.57 (1.83)	11.43 (1.11)	<0.0001 b
**NACA**—M (SD)	3.98 (1.29)	4.46 (1.41)	4.10 (1.00)	5.79 (1.65)	4.06 (1.10)	3.37 (1.16)	<0.0001 b
**Patient Death**—*n* (%)	113 (2.23)	375 (8.02)	2 (0.05)	1236 (48.41)	39 (1.68)	0 (0.00)	<0.0001 a

a—Chi^2^ test; b—Kruskal-Wallis test; GCS—Glasgow Coma Scale; RTS—Revised Trauma Score; NACA—National Advisory Committee for Aeronautics; ACS—acute coronary syndromes; SCA—sudden cardiac arrest.

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
