# Peer review of "Helicopter Emergency Medical Service (HEMS) Response in Rural Areas in Poland: Retrospective Study"

_ijerph, 2019, doi:10.3390/ijerph16091532_

Round 1
Reviewer 1 Report
The authors have focused on an important healthcare intervention related to transport in rural areas of Poland. Few suggestions to consider:
Introduction:
The rationale for conducting such profile analyses of patient characteristics and diagnoses should have been explained further, in terms of policy implications.
Methods:
What was the purpose of doing group analyses as mentioned under data analyses? if that's one of the objectives, need to mention details of rationale in Intro and in Objectives of this study.
Results:
What does 'ambulance support' in table 2 indicate?
What does p value mean in Table 3? Which one was the baseline comparison group, such as for age groups, seasons, RTS, NACA, patient death
Page-7, line: 185: What does the number within bracket indicate? How it was calculated? It needs to be clarified please. Why ORs with 95% CIs were not used to determine the risk/associations?
Discussion:
What would be the explanations for differences in groups, such as gender, or seasons of the year, or values in the scale etc?
Author Response
Dear Reviewer,
The authors would like to thank the Reviewer for many valid comments and suggestions. We have revised our paper as requested. Please find a point-by-point response to the Reviewer’s comments below. The paper has been proofread again by a native speaker of English in a professional translation agency.
The authors have focused on an important healthcare intervention related to transport in rural areas of Poland. Few suggestions to consider:
Introduction:
The rationale for conducting such profile analyses of patient characteristics and diagnoses should have been explained further, in terms of policy implications.
Thank you for the comment. The introduction has been changed according to the suggestion.
Methods:
What was the purpose of doing group analyses as mentioned under data analyses? if that's one of the objectives, need to mention details of rationale in Intro and in Objectives of this study.
Group analysis was not used in the study.
Results:
What does 'ambulance support' in table 2 indicate?
Thank you for your valid suggestion – „ambulance support” means a situation when HEMS crew was called to assist the ground teams. „Ambulance support” has been changed to „Assist the ground team” in Table 2.
What does p value mean in Table 3? Which one was the baseline comparison group, such as for age groups, seasons, RTS, NACA, patient death
In Table 3 p-value pertains to the Chi-square Test in the following variables: gender, age (ranges), seasons of the year, the GCS scale and patient deaths, and in the case of variables, where the mean and standard deviation (age, the RTS score and the NACA score) were presented p-value pertains to the Kruskal-Wallis Test. A legend has been added to Table 3 (a and b) and the table description has been modified according to the used tests. In connection to the above, no comparison groups were distinguished.
Page-7, line: 185: What does the number within bracket indicate? How it was calculated? It needs to be clarified please. Why ORs with 95% CIs were not used to determine the risk/associations?
Thank you for pointing that out. It was an error, which has already been corrected. A legend has been added to Table 4 (a and b) and the table description has been modified according to the used tests. Odds ratio (OR with 95% CIs) was not used due to the characteristics of the examined group – only sick persons, no comparison group. In this study the authors concentrated only on one main health issue in a given patient. It cannot be determined if it was the only issue or if there were more problems. That is why, OR was not used, in order not to make a mistake and present incorrect data.
Discussion:
What would be the explanations for differences in groups, such as gender, or seasons of the year, or values in the scale etc?
Thank you for your suggestion. In our opinion the discussion reflects the presented results, as no comparison groups were distinguished.
The conclusions were changed as suggested.

Reviewer 2 Report
Dear authors,
Congratulations for a very good publication on medical radar using helicopters in rural areas in Poland. The issue is up-to-date, because it is known that the countryside is often located far from hospitals, and help must be given sometimes in a few minutes after the incident. The authors examined all cases throughout the country in the period of 8 years, according to my knowledge since the introduction in Poland of modern helicopters type EC135, enabling faster and more efficient work.
I have a few comments and possible ideas on how to improve this work, or future research in this field:
Maybe it would be an interesting presentation on the charts an operational range and the number of interventions depending on the center.
Are you able to give an exact number of pediatric interventions?Perhaps it would be interesting to present in your study health problems in children. Especially in rural areas in the summer, children are exposed to a lot of unhealthy, often life threatening.
Are you known a circumstance in which accidents have occurred, especially those with head injuries? Can this information be used to plan prevention in rural areas?
How do you describe interventions in Poland, do you have an electronic system or a national register?
Author Response
Dear Reviewer,
The authors would like to thank the Reviewer for the devoted time and for appreciating our manuscript.
The paper has been proofread again by a native speaker of English in a professional translation agency.
Please find a point-by-point response to the Reviewer’s comments below.
Congratulations for a very good publication on medical radar using helicopters in rural areas in Poland. The issue is up-to-date, because it is known that the countryside is often located far from hospitals, and help must be given sometimes in a few minutes after the incident. The authors examined all cases throughout the country in the period of 8 years, according to my knowledge since the introduction in Poland of modern helicopters type EC135, enabling faster and more efficient work.
I have a few comments and possible ideas on how to improve this work, or future research in this field:
Maybe it would be an interesting presentation on the charts an operational range and the number of interventions depending on the center.
Thank you for your suggestion. We presented the information about HEMS in Poland in the introduction. In our opinion the characteristic of HEMS in Poland is not the main purpose of the manuscript.
Are you able to give an exact number of pediatric interventions? Perhaps it would be interesting to present in your study health problems in children. Especially in rural areas in the summer, children are exposed to a lot of unhealthy, often life threatening.
One more time thank you for your valid comment. We agree that health problems in children are very important. As you noticed, especially in rural areas in the summer, children are exposed to a lot of unhealthy, often life-threatening situations. We are currently working on the project concerning paediatric interventions.
Are you known a circumstance in which accidents have occurred, especially those with head injuries? Can this information be used to plan prevention in rural areas?
Thank you for the suggestion. In this case we do not present this information as the data is not sufficient and we were not able to obtain such information.
How do you describe interventions in Poland, do you have an electronic system or a national register?
The interventions in Poland are described by an electronic system maintained by Polish Medical Air Rescue. A national register is being prepared.
